# Demixed shared component analysis of neural population data from multiple brain areas

**Yu Takagi**\*
University of Oxford, UK
yutakagi322@gmail.com

**Steven W. Kennerley**
University College London, UK
s.kennerley@ucl.ac.uk

**Jun-ichiro Hirayama**†
AIST/RIKEN AIP, Japan
junichiro.hirayama@aist.go.jp

**Laurence T. Hunt**†
University of Oxford, UK
laurence.hunt@psych.ox.ac.uk

## Abstract

Recent advances in neuroscience data acquisition allow for the simultaneous recording of large populations of neurons across multiple brain areas while subjects perform complex cognitive tasks. Interpreting these data requires us to index how task-relevant information is shared across brain regions, but this is often confounded by the mixing of different task parameters at the single neuron level. Here, inspired by a method developed for a single brain area, we introduce a new technique for demixing variables across multiple brain areas, called demixed shared component analysis (dSCA). dSCA decomposes population activity into a few components, such that the shared components capture the maximum amount of shared information across brain regions while also depending on relevant task parameters. This yields interpretable components that express which variables are shared between different brain regions and when this information is shared across time. To illustrate our method, we reanalyze two datasets recorded during decision-making tasks in rodents and macaques. We find that dSCA provides new insights into the shared computation between different brain areas in these datasets, relating to several different aspects of decision formation.

## 1 Introduction

Recent methodological advances make it possible to record thousands of neurons simultaneously [1]. Although such high-dimensional recording yields insights that are not apparent from studying single neuron activity, analysing population data remains a non-trivial problem because of the heterogeneity of responses and 'mixing' of encoded variables observed in neural data [2]. While traditionally such heterogeneity was discarded by simply averaging across neurons, more recently several dimensionality reduction methods have been developed for neurophysiological data that isolate key features of the population response structure [3]. One popular approach is demixed principal component analysis (dPCA; [4, 5]). dPCA identifies components that explain maximum population variance while also 'demixed' from non-interesting task parameters, in other words depending upon key task parameters. This allows experimenters to combine rich population recordings with equally rich task design, as dPCA isolates low-dimensional components that vary along axes defined by features of the experimental task.

---

Besides the number of neurons recorded, neuroscientists are also experiencing a revolution in the number of brain areas recorded. This is fascinating because the brain is a connected system comprised of functionally specialised areas that interact with each other to produce complex behaviors. Recently, researchers have started to investigate interactions between populations of neurons in distinct areas for motor control [6], visual processing [7] or perceptual decision making [8]. Similar to studies examining population responses in a single brain region, these studies have applied dimensionality reduction methods to examine information shared across regions - including principal component analysis (PCA; [6]), reduced rank regression (RRR; [7]), or canonical correlation analysis (CCA; [8]).

However, while these approaches all yield important insights into cross-regional information sharing, they cannot identify when and what task parameters are shared across regions. This limits our understanding of how information is shared across regions during cognitive tasks. It is known that the task parameters are mixed at the level of single neuron [9] or low-dimensional components obtained by standard dimensionality reduction methods [10]. Thus, it is important to properly demix each of closely related but distinct task parameters, for example, those encoding decision input, choice formation and motor output during decision making tasks [11, 12]. It is also important to identify precise timing of information sharing because relevant cross-areal information sharing may only occur at any specific timing during the entire task-related processing.

Here, inspired by the approach taken by dPCA, we propose a method for identifying shared components across two areas that is specific to a task parameter of interest in a time-resolved manner. The key idea is to 'marginalize' neural population activity in a single area to demix a specific task parameter of interest, while maximizing the information shared by the two areas with a time lag. We call this procedure demixed shared component analysis (dSCA). [3]

Our contributions are summarized as follows:

1. We briefly review previous methods, emphasizing that they cannot identify what and when task parameters are shared across brain regions.

2. Motivated by this fact, we propose dSCA to isolate the contribution of a task parameter of interest to the shared information between different brain areas. Using a simulation analysis in Sect. 3 we show that dSCA finds shared components that are demixed into specific task parameters of interest in a time-resolved manner, while previous CCA or RRR fail.

3. We reanalyze two previously published decision making datasets [8, 13] and show that dSCA captures shared components among different brain areas that is specific to task parameters such as decision input, stimulus valuation, attentional reorienting and choice formation.

## 2 Related work

**Interaction of populations of neurons across different brain areas.** Early studies investigated interactions of different brain areas in different scales: pairs of single neurons (e.g. [14]), populations of neurons in one area and a single neuron in another (e.g. [15]), neurons in one area and the local field potential (LFP) in another (e.g. [16]), and LFPs in different areas (e.g. [17]). In recent years, some researchers have started to investigate interaction of neuronal populations between brain areas [6, 7, 8]. They commonly applied linear dimensionality reduction methods to study the relationship between neural populations in different areas (Figure 1c, top). For example, Kaufman et al. (2014) [6] applied PCA to each area separately, then regressed from one area to the other area (an approach sometimes referred to as principal component regression). However, the components obtained by PCA that explain maximal variance separately in each area will not necessarily align with those that would explain maximal shared information between the two areas.

More recent studies have used alternative techniques such as RRR [7] and CCA [8, 18] (Figure 1d, top) to directly find low-dimensional latent components from two populations of neurons (Figure 1e, top), optimized for identifying interaction between them. In particular, Steinmetz et al. (2019) [8] also proposed a time-resolved approach, by applying CCA exhaustively on all possible pairs of time points/windows between one area and the other area activities (Figure 1f, top). The method, called joint peri-event canonical correlation (jPECC) analysis, identifies when cross-regional interaction

---

occurs from one region to another. However, none of the above approaches provide results that are demixed by experimental conditions. This means that none of these analyses could identify (*'what'*) information was being shared across regions.

**Demixing task parameters.** It is well known that neurons have mixed selectivity, where a neuron's firing rate reflects more than one task parameter [9]. This is still true for the component after applying standard dimensionality reduction method such as PCA or non-negative matrix factorization [10]. It is a critical problem for researchers who are interested in cognitive processing during a complex experiment, because different types of computations run simultaneously in the brain, and it is important to be able to dissociate computations pertaining to one task parameter from the others. Several approaches have been proposed (e.g., [4, 5, 10]). Among such approaches, dPCA [4, 5] is one of the most popular methods because of its simplicity. dPCA combines regression with dimensionality reduction to demix task parameters of interest. However, this demixing is performed on neurons from a single region at a time, rather than capturing shared information across regions.

## 3 Demixed shared component analysis (dSCA)

We begin with a typical neuroscience experiment that motivated us to develop dSCA. In the experiment, animals are trained on a set of stimuli to make decisions in order to maximize total rewards (Figure 1a, top). For example, animals earn rewards at the end of each task trial if they choose an appropriate action (*Decision*) depending on the visual stimulus (*Stimulus*) presented after a fixation period. Each trial is thus labelled with the two task parameters, Decision and Stimulus, both discrete-valued. Single neuron activities are measured during a series of task trials using an implanted electrode array or probe in the brain. In recent years, such multielectrode technologies have been used for recording populations of neurons simultaneously across multiple brain regions (Figure 1b). In some instances, however, data from different electrodes may be recorded in different sessions, and a 'pseudopopulation' is reconstructed by first averaging across task parameters and then concatenating neurons across recording sessions. We will describe this procedure in detail later.

Our goal is to investigate both the content (*'what'*) and timing (*'when'*) of task-related information sharing among multiple brain areas based upon the measured activities of neuronal populations. We first assume that the entire population was measured simultaneously from area '$X$' and area '$Y$' (but note that our approach generalises to non-simultaneous pseudopopulation recordings below). For each area, we thus observe $M \times T \times N$ arrays of firing rates, where $M$ denote numbers of neurons in the area, $T$ denotes the number of trials, and $N$ denotes the length of timeseries of a particular time window of interest (e.g. 0-500ms after the stimulus onset).

Although existing approaches for cross-areal interaction analysis (see Related Work) can detect low-dimensional representations of shared information between different brain areas, they cannot give insights into the type of information in relation to the task parameters of interest. This is because these approaches use only the data (firing rate) matrices without taking into account how task parameters cause changes in these matrices. As a result, obtained components from these approaches are in general mixed in terms of the task parameters (Figure 1e, top).

This is problematic if we want to study how task-relevant information is passed between brain regions as a cognitive process unfolds. Consider, for example, a decision task in which sensory information is passed in a bottom up sweep from lower to higher cortical areas, but the categorical choice emerges in a distributed fashion across multiple layers in the cortical hierarchy (e.g. [19, 20]; see Figure 1c for simple schematic). Applying standard methods to this data will not differentiate between sensory input causing covariation between brain regions' activity on the one hand, and the emergence of the decision process on the other.

To overcome the problem of what information is shared between regions, we propose to combine the idea of demixing [5] with cross-areal interaction analysis using CCA/RRR. We use demixed Shared Component Analysis (dSCA) to refer to a family of methods that combine these two principles. We focus on RRR for ease of exposition, but note that RRR reduces to CCA if the target matrix is whitened [21] and thus the framework can unify both techniques.

First, recall that standard least-squares RRR minimizes the following:

$$L_{\mathrm{RRR}} = \|\mathbf{Y} - \mathbf{W}\mathbf{X}\|^2$$

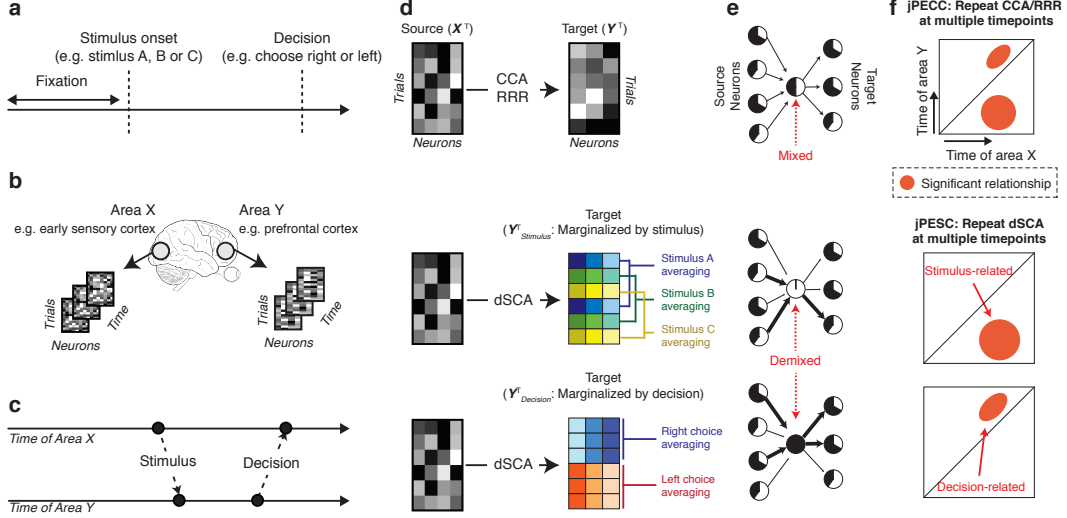

Figure 1: **dSCA can identify content and timing of task-related information sharing among multiple brain areas. a** Schematic of typical timeline of a trial in a decision making experiment in neuroscience. **b** The data are represented as sequences of populations of neurons matrices of area $X$ (e.g. early sensory cortex) and area $Y$ (e.g. prefrontal cortex). **c** area $X$ (top) and area $Y$ (bottom) communicate task-related information each other. **d** Schematic of the analysis for two populations of neurons from different brain areas, $X$ and $Y$. Conventional approaches (CCA or RRR) estimate relationship between source and target populations of neurons without taking task parameters into account (top). In contrast, dSCA applies marginalization to the target matrix by averaging across specific task-parameter of interests (middle: marginalization for stimulus; bottom: marginalization for decision). In the marginalization procedure, for each neuron and each task parameter, trials having the same level of the task parameter have the same values (average across the same-level trials). Cells with the same colours indicate the same values. **e** Schematic of the obtained components from CCA/RRR (top) and dSCA (middle and bottom). Each circle indicates a neuron or a component, and the width of an arrow indicates estimated weights. The component obtained from CCA/RRR is mixed, whereas the components obtained via dSCA are demixed in terms of the task parameter. **f** Top: Schematic of joint peri-event canonical correlation (jPECC) analysis. Results obtained from CCA/RRR may not dissociate two task parameters from one another. Using dSCA yields joint peri-event shared component (jPESC) for stimulus (middle) and decision (bottom) are dissociated. Coloured areas indicate significant relationship between two areas at the pair of time-points.

where $\mathbf{X}$ is the area $X$'s data matrix of size $M_X \times T$, $\mathbf{Y}$ is the area $Y$'s data matrix of size $M_Y \times T$, and $\mathbf{W}$ is coefficient matrix of size $M_X \times M_Y$, and the rank of $\mathbf{W}$ is constrained not to be greater than $K$ ($< min(M_X, M_Y)$); $M_X$ and $M_Y$ denote their respective numbers of neurons , and the number of columns in $\mathbf{X}$ and $\mathbf{Y}$ equals the number of trials $T$ in our time-resolved setting while it may vary depending on the context. The constrained minimization can be solved using the singular value decomposition:

$$\mathbf{W}_{RRR} = \mathbf{W}_{OLS}\mathbf{V}\mathbf{V}^T$$

where $\mathbf{W}_{OLS}$ is the ordinary least-squares solution and the columns of the $M_X \times K$ matrix $\mathbf{V}$ contain the top $K$ principal components of the optimal linear predictor $\hat{\mathbf{Y}}_{OLS} = \mathbf{X}\mathbf{W}_{OLS}$.

To properly demix the effect of task parameters within CCA/RRR, we now make a key change to the above framework based on the so-called 'marginalization' procedure [5]. The idea is to replace every column of raw target matrix $\mathbf{Y}$ with its conditional expectation, estimated empirically, given the corresponding realization of a specific task parameter of interest (Figure 1d, middle and bottom). For example, in the running example above, there can be 3 possible stimuli and 2 possible decisions. Marginalization to demix 'stimuli' yields $\mathbf{Y}_{Stimulus}$ having identical columns (trials) if they correspond to the same type of stimulus irrespective to the type of decision (Figure 1d, middle). Similarly, marginalization for 'decision' yields $\mathbf{Y}_{Decision}$ with identical columns depending only on

the type of decision for each trial (Figure 1d, bottom). We will generically write the marginalized matrix as $\mathbf{Y}_m$, where $m$ can be any task parameter of interest, such as stimulus or decision, containing $N_m$ possible levels. We can also consider marginalization for the interaction of multiple parameters, e.g. stimulus and decision.

We refer to the resultant analysis framework as dSCA. After solving the CCA/RRR, if $\mathbf{Y}_m$ can significantly be associated with the source activity via the low-rank representations, it indicates that areas $X$ and $Y$ share information that is relevant to the task parameter $m$ of interest. For example, if we are interested in the stimulus-related information sharing between areas $X$ and $Y$, we will marginalize the target matrix by stimulus, thus we use $\mathbf{Y}_{Stimulus}$ as a target matrix. Then, applying RRR as above, we can find an optimal low-rank representation of the source populations of neurons for predicting the target area's variance that is related to the stimulus information (Figure 1e, middle and bottom). Thus, dSCA explicitly takes task parameters into account, which is the crucial difference from related previous methods.

Note that in our framework, we only marginalize the target, with the 'source' matrix $\mathbf{X}$ being left intact. As discussed in [5], the underlying idea is that while the marginalized target can eliminate task-irrelevant variability by marginalization, one can still employ full information in the source populations of neurons. In fact, one may apply marginalization to source matrix as well in addition to the target, which gives a simple variation of dSCA. If the effects of different task parameters on neural populations are independent, the results obtained from the original source matrix and marginalized source matrix are indifferent. However, this reduces the effective sample size rather drastically (as the column pairs in $\mathbf{X}_m$ and $\mathbf{Y}_m$ then contain many duplicates).Therefore, if their effects are not independent, marginalization by non-interesting task parameters may unintentionally diminish the information of the task parameter of interest. Indeed, we empirically found that marginalization of both matrices often leads to less accurate results (see Section 4.1 for simulation analysis).

So far, we have assumed that all the neurons are measured simultaneously, but this is not always the case. Fortunately, when we do not record all neurons simultaneously, we can still apply the same technique by making use of the concept of 'pseudopopulations', as discussed in [5]. For each neuron, we first compute summary statistics for all possible combinations of task parameters, called peri-stimulus time histogram or PSTH. To calculate PSTH, we will average each neuron's firing rate over trials for each possible task parameter, in order to estimate the neuron's time-dependent firing rate. For example, suppose that we have two task parameters, stimulus and decision. In this case, we will average over trials for each stimulus $s$ (out of $S$) and decision $d$ (out of $D$). Specifically, we will use two matrices of $\mathbf{X} \in \mathbb{R}^{M_X \times C}$ and $\mathbf{Y} \in \mathbb{R}^{M_Y \times C}$, where $C = S \times D$.

To address the question of when information is shared between the two regions, we follow the jPECC approach introduced in Steinmetz et al. (2019) [8]. We repeat the dSCA analysis at all possible combinations of time-bins between areas $X$ and $Y$. This yields a $N$-by-$N$ peri-event matrix for each demixed shared component (Figure 1f), which we refer to as the joint peri-event shared components (jPESC). When performing this lagged analysis, we assume that whichever neuronal population is currently earlier in time is the 'source' matrix, and whichever population is later in time is the 'target' matrix.

## 4 Results

**Synthetic data**    To illustrate that dSCA can detect shared components between two areas that is specific to a task parameter of interest, we generated two simulated neuronal population datasets.

Suppose that we simultaneously recorded from two brain areas, $X$ and $Y$, during the experiment in which two task parameters exist: stimulus ($S$) and decision ($D$). The neural population in $X$ is affected by variation in the stimulus, as shown in Figure 1c. Then, the population of neurons in $X$ passes stimulus information to another population of neurons in $Y$ with some time delay. A decision computation then arises in the population of neurons $Y$, such that it begins to encode the categorical choice of the animal, which is passed back to populations of neurons in $X$. We also added independent random noise to populations of neurons in $X$ and $Y$ (see Supplementary Material).

We first conducted jPECC on the simulated data, applying standard CCA and RRR to $X$ and $Y$ with different time point pairs (Figure 2a), as was done in Steinmetz et al. (2019) [8]. Each pixel in the resulting matrix represents the strength of cross-validated correlation between the first pair of

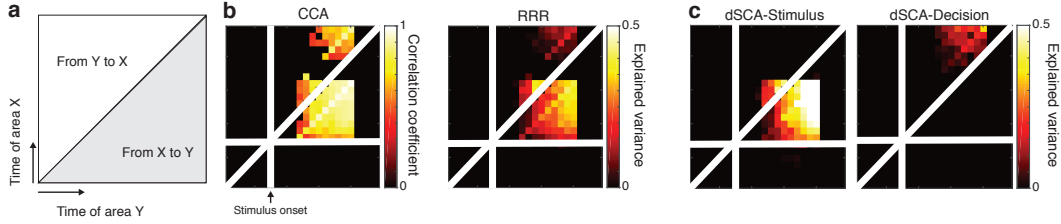

Figure 2: **Simulation demonstrates that dSCA decomposes information sharing between different brain regions into different task-relevant parameters. a** Schematic of joint peri-event analysis. **b** Results obtained from CCA (left) and RRR (right) show similar results. **c** Results obtained from dSCA for stimulus (left) and decision (right).

canonical variables at a given latency for CCA, or explained variance, $-(\mathbf{Y}_m - \hat{\mathbf{Y}}_m)^2/\text{Var}(\mathbf{Y}_m)$, for RRR. Although jPECC can reveal that these two areas share information (significant clusters are shown; $P < 0.05$, cluster-based permutation test, corrected for multiple comparisons; see Supplementary Material for details), it does not tell us what types of information is being shared (Figure 2b). Given that both CCA and RRR provide similar results in the simulation analyses, we will use CCA for the following analyses of real datasets. Note that, if we used explained variance as a metric for CCA, we obtained similar results to using correlation (see Supplementary Figure 1).

We next applied jPESC to $X$ and $Y$, to obtain a similar time-resolved matrix but using dSCA rather than CCA/RRR. We focus on two main task parameters: stimulus ($S$) and decision ($D$). In contrast to CCA, we used the marginalized matrix as the target, rather than the raw population matrix. Figure 2c shows that dSCA can clearly dissociate the shared information related to the distinct task parameters in populations of neurons in area $X$ and $Y$ ($P < 0.05$, cluster-based permutation test, corrected for multiple comparisons). Note that, although we used the raw matrix as the source matrix for dSCA, we could also use the marginalized source matrix (as described in section 2.2). We found that such a version of dSCA in which both $X$ and $Y$ are marginalized provides similar, but less accurate results than the results from using the raw source matrix (see Supplementary Figure 2). Therefore, we used the raw matrix as the source matrix for the following analyses of real datasets.

**Perceptual decision making**   Next, we reanalyzed a perceptual decision making dataset for validating the use of dSCA to demix task parameters of interest. We used the dataset that was published in Steinmetz et al. (2019) [8] . On each trial of the experiment (Figure 3a), visual stimuli of varying coherence could appear on the left side, right side, both sides or neither side. Mice earned a water reward by turning a wheel with their forepaws to indicate which side had the higher coherence. Here, we exclude NoGo trials, where neither stimulus was present and mice should not move.

In the original study, the authors analyzed interactions of neural populations using jPECC among three different subregions: visual cortex, frontal cortex, and midbrain (Figure 3b); see Steinmetz et al. (2019) for precise anatomical locations included in each of these subregions [8]. They applied jPECC to neural activities that were simultaneously recorded at relative to movement onset between visual and frontal cortex (Figure 3c, top), visual cortex and midbrain (Figure 3c, middle), and frontal cortex and midbrain (Figure 3c, bottom). Their results revealed the latency at how information is shared between these subregions, but not which task parameter was being shared. We preprocessed the dataset exactly as in the original study (see Supplementary Material for the details of preprocessing) and our results with standard jPECC replicated their previous findings (Figure 3c; significant clusters are shown; $P < 0.05$, cluster-based permutation test, corrected for multiple comparisons). Note that, for each combination, we analysed data from the three sessions with the largest number of completed trials; this is because there was a substantial variation in terms of the number of completed trials between sessions, and a certain amount of trials are necessary for reliable estimation by dSCA. Qualitatively similar (albeit weaker) results could be obtained from all sessions, including those with fewer trials (Supplementary Figure 4).

To obtain demixed results, we applied dSCA to the same dataset. Here, we focus on two main task parameters: stimulus and decision. Stimulus is defined as the strength of left coherence (1, 0.5, 0.25 or 0) minus strength of right coherence (1, 0.5, 0.25 or 0) for each trial. Decision is defined as the mouse's choice for each trial (Left or Right). In addition to applying dSCA to raw population

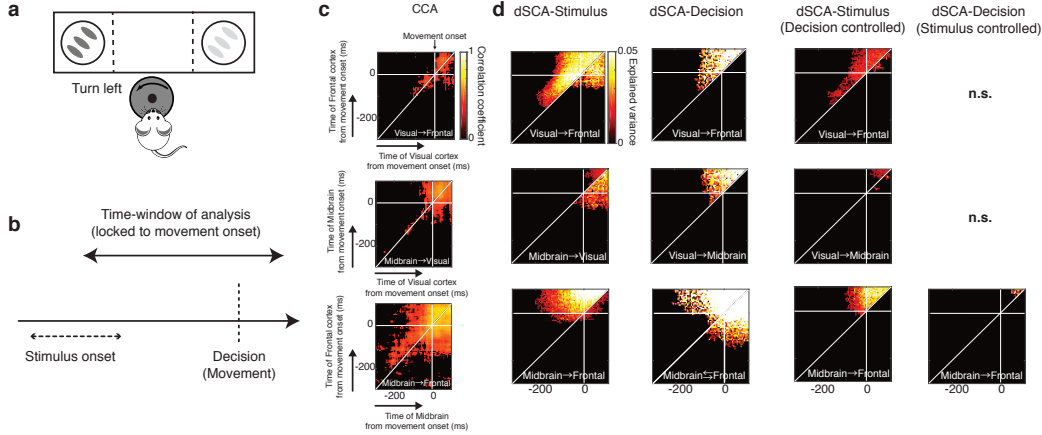

Figure 3: **DSCA reveals strong double-dissociation of information sharing between two task parameters that are not apparent by CCA. a** Schematic of mice turning a wheel to indicate which of two visual gratings had higher contrast. **b** Schematic of task sequence. We focus on the time-window of -300 – 100 ms after decision (movement onset). **c** Results obtained from CCA. Directional arrows indicate significant time lag, whereas bidirectional arrows indicate no significant time lag. **d** Results obtained from dSCA for sensory inputs (1st column), movement direction (2nd column), sensory inputs that controlled movement direction (3rd column), and movement direction that controlled sensory inputs (4th column). n.s. corresponds to no significant cluster.

matrices, we also applied dSCA to matrices after regressing out either stimulus or decision from these matrices, because these two task parameters are correlated to some extent (see Supplementary Material). Figure 3d shows that for all combinations of brain regions, information sharing is clearly decomposed into stimulus- and decision-related components (significant jPESC clusters are shown; $P < 0.05$, cluster-based permutation test, corrected for multiple comparisons). We can also see different types of time lag. For example, between midbrain and frontal cortex (bottom row), the shared stimulus-related component is lagged (occurring earlier in midbrain than in frontal cortex); by contrast, the decision-related component emerged in parallel between them (directional arrows indicate significant time delay, whereas bidirectional arrows indicate non time delay; $P < 0.05$; see Supplementary Material for details of the statistical inference). We confirmed that marginalizing both of the source and target regions provides similar results (see Supplementary Figure 3). This underscores the unique contribution of dSCA in allowing us to observe when and what types of information is shared across different brain areas.

**Economic decision making**   Finally, we applied dSCA to recordings from a macaque monkey performing an attention-guided information search and economic choice task. The data used here were previously published in Hunt et al. (2018) [13]. On each trial, a monkey made an instructed saccade toward a highlighted location to reveal a picture cue. The cue indicated to the monkey either the probability or magnitude of reward that would be available from a subsequent choice towards that spatial location. After 300ms of uninterrupted fixation, cue 1 was covered, and another cue location was highlighted. Full details of the information search and choice structure of the task can be found in Hunt et al. (2018) [13]; here, we only focus on the time when cue 1 was first attended to.

The authors recorded from three prefrontal cortex (PFC) subregions (anterior cingulate cortex [ACC], orbitofrontal cortex [OFC], and dorsolateral prefrontal cortex [DLPFC]), and analyzed each area separately in their original paper. The authors previously analysed these data only within-region, and capitalized on neuronal heterogeneity to assessing population-level encoding of cue value and spatial location, amongst other variables. Although they found a strong dissociation between the three PFC subregions in the degree of population encoding, all subregions had some encoding of both value and space. We therefore sought to use dSCA to identify how value and space information was shared between the three subregions, timelocked to the presentation of the stimulus.

As with the previous analyses, we first applied CCA to the combinations of ACC, OFC and DLPFC. We preprocessed the dataset exactly the same way as the original study. Note that, unlike the dataset

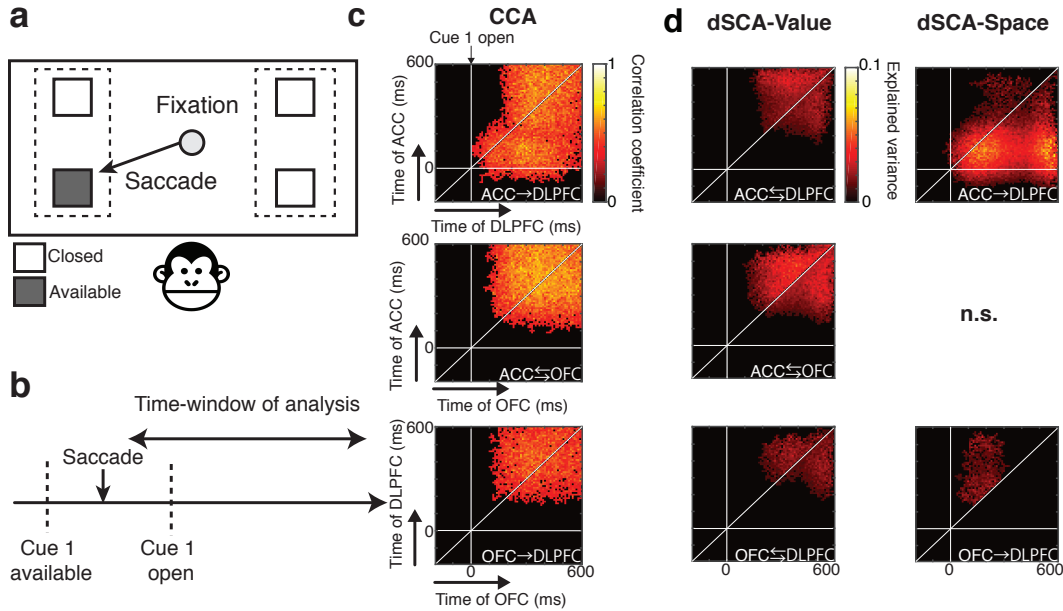

Figure 4: **dSCA reveals that value-related information is strongly shared between OFC-ACC, emerging simultaneously across both subregions, whereas space-related information is shared between DLPFC and the other regions with a time lag. a** Schematic of task. Monkey chose between a left and right option (dotted rectangles), after sequentially sampling cues that revealed reward probability and magnitude. Gray squares indicate locations available for sampling. **b** Schematic of task sequence. We focus on the time-window of -200 – 600 ms after cue 1 onset. **c** jPECC results obtained from CCA. Directional arrows indicate significant time lag, whereas bidirectional arrows indicate no significant time lag. **d** jPESC obtained from dSCA for value (left) and space (right). n.s. corresponds to a no significant cluster.

published in Steinmetz et al. (2019), we performed the analysis on 'pseudopopulations' because not all neurons were simultaneously recorded (see Supplementary Material). Figure 4c applies standard jPECC to the data, and shows that all pairs of PFC subregions shared information after cue 1 onset (significant clusters are shown; $P < 0.05$, cluster-based permutation test, corrected for multiple comparisons). However, again, these results cannot tell us what type of information is being shared.

To obtain demixed results, we next applied dSCA to these pairs of PFC subregions. We focused on two main task parameters in the task: space and attended-value. Space is defined as the direction of cue 1 (Left Option or Right Option). Attended-value is defined as the magnitude (value or probability) of cue 1 (1, 2, 3, 4 or 5). As spatial position and value are orthogonal by task design, there is no need to regress these out of the data prior to marginalisation as in Figure 3.

Figure 4d shows that, again, the jPESC with dSCA captures several characteristics that are not apparent from the jPECC with CCA. Between ACC and DLPFC, value- and space-related shared components was strongly dissociated after cue 1 onset ($P < 0.05$, corrected for multiple comparisons, cluster-based permutation test, only significant clusters are shown; see Supplementary Material for details); between ACC and OFC, we can see the strongest value-related shared components among all combinations, whereas no space-related shared components emerged; space-related components in ACC/OFC just after stimulus onset were sustained in DLPFC, whereas value-related computation emerged relatively in parallel later ($P < 0.05$; see Supplementary Material for details). We confirmed that marginalizing both of the source and target regions provides similar results (see Supplementary Figure 3).

In summary, compared to previous methods, dSCA can provide us with insight into how information is shared in terms across brain regions, in terms of a specific task parameter of interest.

# 5 Conclusion and discussion

We proposed dSCA, a new technique for analysing populations of neurons obtained from different brain areas. Unlike previous methods, dSCA decomposes population activities into a few components to find a low-rank approximation that maximizes the information shared by multiple brain areas with a time lag, while also depending on a relevant task parameter. We demonstrated that dSCA can reveal task specific shared components that are overlooked by conventional approaches using simulation and two previously published neuroscience datasets. We believe dSCA will be useful for neuroscientists who will have a large amount of data from different brain areas during complex cognitive experiments.

Our method has several limitations. First, dSCA assumes that task-related communication is linearly represented. It makes dSCA simple and exactly solvable, and a linear method is popular in neuroscience because of its interpretability and less computational demand (see Supplementary Figure 5 for the interpretation of the shared component in the simulation and real datasets). We believe our method is a good starting point for practitioners and methodological exploration. Second, even if we find a significant relationship between two regions via dSCA, this does not guarantee that they are communicating directly because there is always the possibility of a third region sharing information with both source and target regions. Despite this limitation, our method is an important starting point for subsequent interventional studies that more explicitly test task-related communication in a causal manner.

Future research topics include: (i) extending dSCA to deal with more than two brain areas; (ii) investigating the characteristics of components obtained from dSCA at the level of single-trial, for example, what is the behavioural difference between trials in which value-related information is shared between ACC-OFC and trials in which value-related information is not shared between them; and (iii) applying dSCA to human neuroimaging data measured by magnetoencephalography, or electrical potentials measured by electrocorticography.

## Broader Impact

Although several studies have investigated communication between populations of neurons, task-related communication has been ignored. This is of fundamental importance in neuroscience, and we show that it can be achieved simply by extending the previous method. We believe our methods will be beneficial to the neuroscientists who will investigate interaction among multiple brain areas in terms of specific task parameter of interest.

## Acknowledgments and Disclosure of Funding

YT was supported by Grants-in-Aid for Scientific Research on Innovative Areas from the JSPS (23118001, 23118002) and Uehara Memorial Foundation. JH was supported by JSPS KAKENHI (18KK0284). LH was supported by a Sir Henry Dale Fellowship from Wellcome and the Royal Society (208789/Z/17/Z).

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
