[Supplementary Material]

# Supplementary Marterial:
# Demixed shared component analysis of neural population data from multiple brain areas

**Yu Takagi**        **Steven W. Kennerley**        **Jun-ichiro Hirayama**        **Laurence T. Hunt**

## 1   Simulation analyses

### 1.1   Dataset

We generated sequences of neuronal populations in areas $X$ (e.g. early sensory cortex) and $Y$ (e.g. prefrontal cortex). We set the number of time steps to 24, and the number of neurons in areas $X$ and $Y$ to 10 and 9, respectively. At each time step $t$, baseline activities for areas $X$ and $Y$ were drawn randomly from a Gaussian distribution $\mathcal{N}(0, 1)$. Each trial was labelled with stimulus and decision. There were 5 possible stimuli and 3 possible decisions, thus $3 \times 5 = 15$ possible combinations of labels. For each combination, we generated 20 trials, resulting in 300 trials in total. Stimulus onset occurred at time step 6. Noises were added to each neuron for each trial independently from a Gaussian distribution $\mathcal{N}(0, \frac{1}{3.5})$.

Neurons in areas $X$ and $Y$ were affected by the stimulus and decision, and communicated with each other as follows. First, neurons in area $X$ were affected by the stimulus for three time steps from the time of stimulus onset. The magnitudes of the effects were randomly determined for each neuron, were linearly correlated with the level of the stimulus, and were linearly increased across time. Neurons in area $X$ passed the stimulus-related information to the neurons in area $Y$ via a random projection matrix after two time steps from the time when neurons in area $X$ started to process stimulus-related computation. After area $Y$ received the stimulus-related input from area $X$, neurons in area $Y$ started to compute the decision. As with the stimulus-related computation in area $X$, the magnitude of the effects were also randomly determined for each neuron, were linearly correlated with the level of the decision, and were linearly increased across time. Area $Y$ then passed decision-related information back to area $X$ via another random projection matrix after two time steps from the time when decision-related computation in $Y$ emerged.

### 1.2   Analyses

With the setting described above, we obtained a $10 neurons \times 300 trials \times 24 timesteps$ array for area $X$, and $9 neurons \times 300 trials \times 24 timesteps$ array for area $Y$. To investigate the relationship between areas $X$ and $Y$ in a time-resolved manner, we then constructed two matrices $\mathbf{X}$ and $\mathbf{Y}$, where $\mathbf{X}$ is the area $X$'s data matrix of size $10 \times 300$ at time $t_X$, and $\mathbf{Y}$ is the area $Y$'s data matrix of size $9 \times 300$ at time $t_Y$. We assume that whichever neuronal population is currently earlier in time is the 'source' matrix, and whichever population is later in time is the 'target' matrix . We ignored the cases where $t_X == t_Y$.

To separate the data into training and testing sets, for each label combination, we held out one random trial as test trial. Thus the number of test set trials is $3 decisions \times 5 stimuli = 15$. We then applied jPECC (with CCA/RRR) or jPESC (with dSCA) to training set as follows. We then applied obtained transformation matrices to test set. We repeated this procedure fifteen times for different train-test splits, and averaged the results.

jPECC with CCA/RRR was performed on the training set, L2 regularized using $\lambda = 0.05$. For CCA, the test set trials were projected onto a canonical dimension, and the Pearson correlation coefficient was computed between projected test set trials in the source and target areas. For RRR, we predicted values of the test set target matrix from the test set source matrix, using transformation matrices estimated by training set. We then computed explained variance between predicted and actual test set target matrix. We used the top low-dimensional component both for CCA and RRR.

As with the jPECC with CCA/RRR, jPESC with dSCA was performed on the training set, L2 regularized using $\lambda = 0.05$. However, here we applied marginalization to the target matrix. We used stimulus- and decision-marginalized target matrices for dSCA separately. We predicted values of the marginalized target matrix from the source matrix in the test set, using transformation matrices estimated by the training set. We then computed explained variance between predicted and actual target matrices. We used the top low-dimensional component.

### 1.3 Statistical inference

To test whether areas of high correlation/explained-variance between two brain regions were significantly larger than would be expected by chance, we used a cluster-based permutation test [1]. For jPECC with CCA, we identified clusters in the correlation map that were larger than a cluster-forming threshold (set at $r > 0.4$). For jPECC with RRR and jPESC with dSCA, we identified clusters in the explained variance map that were larger than a cluster-forming threshold (set at $R > -0.99$). We then permuted trials in one brain area to recompute jPECC/jPESC, and we identified clusters that exceeded the cluster-forming threshold in the permuted data. Note that, we permuted trials but not distorted temporal structure across time steps. For each of the 100 permutations, we stored the size of the largest cluster. This procedure provided a null distribution of maximum cluster sizes that would be expected by chance. We used 95th percentile of this null distribution as a threshold for deeming whether the cluster sizes observed in the data were significant, at $P < 0.05$ (corrected for multiple comparisons across all pairs of timepoints).

## 2 Perceptual decision making task (Steinmetz et al., 2019)

### 2.1 Dataset

The authors trained mice to perform visual discrimination task. During each recording session, the authors simultaneously recorded from hundreds of neurons in multiple regions using Neuropixels [2, 3] probe ($n = 92$ probe insertions over 39 sessions in 10 mice). The details of data acquisition and preprocessing are described in [4]. Datasets is obtained from `https://figshare.com/articles/Dataset_from_Steinmetz_et_al_2019/9598406` and codes for preprocessing is obtained from `https://github.com/nsteinme/steinmetz-et-al-2019`, both were published by the authors.

In their original paper, they applied jPECC with CCA to neural activities that were recorded at relative to movement (-300 to 100 ms) between visual and frontal cortex (15 sessions), visual cortex and midbrain (10 sessions), and frontal cortex and midbrain (9 sessions) (see [4] for precise anatomical locations included in each of these subregions). For each combination, we analysed data from the three sessions with the largest number of completed trials; this is because there was a substantial variation in terms of the number of completed trials between sessions, and we found empirically that a certain amount of trials are necessary for reliable estimation by dSCA.

If two task parameters are orthogonalized by experimental design, we do not need to do any procedure before marginalization (as in economic decision making task). However, if two task parameters are correlated by design (as in this task), to focus on a task parameter, we need to regress the other task parameters out from neural data before marginalization. Therefore, we also prepared the neural data that were regressed out Stimulus or Decision information. These regressed-out matrices were also used for dSCA.

### 2.2 Analyses

We applied jPECC with CCA and jPESC with dSCA. All settings are the same as the simulation analyses except for the following:

- Before applying jPECC with CCA and jPESC with dSCA, we applied PCA to both source and target matrices as was done in [4], across time points and trials to reduce population activity to ten dimensions.
- For jPESC with dSCA, marginalization was performed two main task parameters: stimulus and decision (see main manuscript for the definition). We also applied dSCA to matrices after regressing out either stimulus or decision from these matrices. We confirmed that applying PCA before applying jPESC with dSCA did not substantially change the results (data not shown).
- We repeated the cross-validation procedure ten times, averaging the results.

### 2.3 Statistical inference

We used the same cluster-based permutation test procedure as described above for the simulation analyses.

To quantify lead–lag relationships, we computed the asymmetry index by calculating the number of significant timepoints included in the identified cluster from the above permutation procedure. We calculated these numbers in right and left half of the obtained jPECC and jPESC matrices separately, and then subtracted right from left. This procedure provided a null distribution of the difference in the number of significant timepoints between left half and right half that would be expected by chance. We used the 95th percentile of this null distribution as a threshold for deeming whether the difference of number of significant timepoints between the left half and right half observed in the data were significant, at $P < 0.05$.

## 3 Economic decision making task (Hunt et al., 2018)

### 3.1 Dataset

The authors recorded neuronal activity in the macaque OFC, ACC and DLPFC during sequential attention-guided information search and choice. During a typical recording session, 8–24 electrodes were lowered bilaterally into multiple target regions. We used the data from monkey coded 'F'. The details of data acquisition and preprocessing are described in [1]. Dataset and codes for preprocessing are obtained from `http://crcns.org/data-sets/pfc/pfc-7` that was published by the authors.

In this experiment, neural recordings were obtained in multiple sessions, so most of the neurons were not recorded simultaneously. We therefore used 'pseudopopulation' matrices by averaging averaged across task parameters to identify each neuron's response to experimental variables. This allowed us to collapse data across sessions.

### 3.2 Analyses

We applied jPECC with CCA and jPESC with dSCA, as per the previous analyses. All procedures are the same as the simulation analyses except for the following settings:

- For jPECC with CCA and jPESC with dSCA, we applied PCA to both source and target matrices as was done in [4], across time points and trials to reduce population activity to eight dimensions.
- For jPESC with dSCA, marginalization was performed on two main task parameters in the task: space and attended-value (see main manuscript for the definition). We confirmed that applying PCA before applying jPESC with dSCA did not substantially change the results (data not shown).
- For cross-validation in the pseudopopulation setting, to separate the data into training and testing sets, we followed the procedure proposed in [5]. We first held out one random trial for each neuron in each combination of task parameters, i.e. space and attended-value, as a set of test pseudopopulations $X_{test}$ and $Y_{test}$. Because there are 5 possible stimuli and 2 possible spaces, the dimensions of $X_{test}$ and $Y_{test}$ is $5 \times 2 = 10$. Remaining trials were averaged to form a training sets of $X_{train}$ and $Y_{train}$. Note that test and training sets ($X_{test}$ and $X_{train}$, or $Y_{test}$ and $Y_{train}$) have the same dimensions of 10. We repeated this cross-validation procedure 20 times and averaged the results.

### 3.3 Statistical inference

We used the same cluster-based permutation test procedure as described above for the simulation analyses. We also used the same permutation test procedure for determining lead–lag relationships as described above for the perceptual decision making task.

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

Figure 1: **Different metrics/methods provide similar results.** CCA can identify two clusters when accuracy is defined as correlation coefficient (left) or explained variance (middle). RRR (right) also provides similar results

Figure 2: **Simulation results obtained from non-marginalized source matrix (top row) much clearly captured the interaction between two areas compared to the results obtained from marginalized source matrix (bottom row) that captured the smaller size of clusters.**

Figure 3: **Marginalized source matrix for (a) perceptual decision-making task and (b) economic decision-making task provide qualitatively similar results.**

Figure 4: **For data from mice (Steinmetz et al., 2019), qualitatively similar (albeit weaker) results could be obtained from all sessions, including those with fewer trials. a** jPECC results obtained from CCA are shown. For visualization purposes, we used arbitrary cluster-forming threshold: r > 0.3 and size-of-cluster > 5. **b** jPESC results obtained from dSCA are shown. For visualization purposes, we used arbitrary cluster-forming thresholds: for three regressors (stimulus, decision, and stimulus that was controlled by decision), we used explained-variance > -0.99 and size-of-cluster > 10; for the regressor of decision that was controlled by stimulus, we used a less cluster-forming threshold (explained-variance > -1.01 and size-of-cluster > 10) due to the lower effect size.

Figure 5: **Interpretation of the shared components by investigating the contribution of each neuron.** We visualize contribution (absolute weights) to the first shared component for each neuron in the **a** simulation analysis and **b** economic decision making task. For economic decision ask, maximum absolute weights across time are visualized for each neuron. The result is highly interpretable: for example in the simulation analysis, neurons that contributed to Stimulus have higher contribution for stimulus-related communication than the others (left), whereas neurons that contributed to Decision have higher contribution for decision-related communication than the others (right).