[Reviews · NeurIPS 2020]

Review 1

Summary and Contributions: The authors propose a new data analysis method for recordings of multiple brain areas. The method is essentially a form of reduced rank regression from one brain area to another, in which the regression targets are chosen in a way that isolates the representation of a particular variable of interest (an approach that was used for demixed PCA; refs. [4] and [5]). This allows one to ask the question: can one use the activity of brain area X as a basis to explain area Y's representation of the task variable?

Strengths: The proposed method allows the user to examine correlations between brain areas that are related to the representation of particular variables in the target area, rather than arbitrary correlations. This could be useful for neuroscience practitioners that record from multiple brain areas. The demonstartion of the method applied to real data is a strength.

Weaknesses: My major concern with this paper is the interpretation of the results. The manuscript is presented as a method to identify "task-related communication" among brain areas. However, it is not clear that it accomplishes this. In particular, wouldn't two brain areas that encode a particular stimulus, but do not necessarily "communicate" with one another (perhaps they both receive input from a primary sensory area, for example), exhibit a significant relationship simply because one can be regressed against one another? The fact that stimulus representations are averaged (Fig. 1b, center) when constructing the regression targets means that there is no opportunity for trial-to-trial fluctuations in the first area to influence the correlation. This seems to argue against interpreting the results as "task-related communication." Instead, it may simply mean that both areas represent the variable. Related to the above, the discussion on lines 161--168 is not clear. Why is the "full information" in the source population useful if the target population is marginalized? The extra within-stimulus variability in the source population should not be useful in explaining a target that has no fluctuations conditioned on the stimulus.

Correctness: The results appear correct. However, the phrasing of the claim about "task-related communication," as described above, does not appear to be justified.

Clarity: The manuscript is well-written, but should give more time to the issues of causality and interpretation.

Relation to Prior Work: The results are strongly influenced by demixed PCA (ref. [5]) and other recent results, and these are well cited.

Reproducibility: Yes

Additional Feedback: I appreciate the author's response and increased my score.


Review 2

Summary and Contributions: The paper proposed a demixing approach to identify task-relevant shared information between brain areas. They defined a known demixing approach in a way to associate a component which can be interpreted as the part of variance which is shared between different brain areas.

Strengths: This method can have broad applications in simplifying some of the complex structures in big neuroscience multi area datasets. It is based on a known and simple approach and can have broad applications in neuroscience and for the NeurIPS audience.

Weaknesses: The method lacks enough novelty and originality as a new computational approach. Deciding a prior structure on the data, here breaking the neural population into subsets of neurons as each area, and marginalizing over task variables as a way to separate their individual contribution doesn’t seem to be more than a particular design for an existing computational approach. Moreover, the approach assumes some assumptions in order to separate the independent factors which might not be applicable to data. For example, the marginalization over task variables means that they ignore the possible existence of correlations between task variables and also the fact that different areas might share or communicate the interaction of task variables.

Correctness: The method is correct within its assumptions but the validity of these assumptions in real data should have been discussed better. Something which is not very clear is the interpretability of the shared component and how the content of the shared information can be understood.

Clarity: The paper is written clearly. It could be beneficial to elaborate better the limitations of the method both in terms of assumptions taken in the model and also the computational limitation on sample size and possible degeneracy in the solutions.

Relation to Prior Work: I think the differentiation and advantages of this work with other approaches such as dPCA and even NMF specially in terms of real world questins could be better explained.

Reproducibility: Yes

Additional Feedback:


Review 3

Summary and Contributions: The authors present an extension of demixed principal component analysis to the problem of analyzing how information is shared across multiple brain areas. The result is a dimensionality reduction method that aligns the low-D components that relate the activity in two regions with experimental variables of interest.

Strengths: The methods and results are generally clear and grounded, and it provides a method that targets an important, general problem in neuroscience. This extension of dPCA and CCA is of potential interest to many neuroscientists using various large-scale brain recording methods.

Weaknesses: One component that felt missing from the paper is a clear outline of the limitations of the approach. What assumptions are necessary for the data (how neurons mix signals) and experimental design? How robust is the method to violations of these assumptions? For example, line 292 says that dSCA “maximizes the information shared by multiple brain areas”, but this isn’t information in the Shannon sense. It’s restricted to linear/squared-error assumptions. These should be discussed so that users can make better judgements about whether or not dSCA is an appropriate tool for their datasets & questions.

Correctness: The methods and claims were correct. The scaling on the explained variance points was unclear to me (Fig 2b-c). Why is the scale all negative (-1 to -0.5)? UPDATE: The authors' response has addressed the scaling.

Clarity: The paper clearly presented the methods and results. Minor: A quick definition of what “demix” means would be good in the introduction. It’s not explicitly there right now. Line 276-278: I didn’t follow this sentence, or what this should tell practitioners when applying the marginalization procedure. UPDATE: With the limited space in the response, I still am a bit uncertain about what the exact regression procedure suggested by the authors is. However, it sounds simple enough that I believe that it'll be covered in a revised manuscript.

Relation to Prior Work: The introduction links the method to existing approaches (e.g., dPCA, CCA) and explains what the current paper contributes.

Reproducibility: Yes

Additional Feedback: Minor suggestions: Paragraph on line 74: another related method that could be cited if the authors think necessary is dynamic kernel CCA: Rodu, Jordan, et al. "Detecting multivariate cross-correlation between brain regions." Journal of neurophysiology 120.4 (2018): 1962-1972. Line 83: “where a neuron” -> “where a neuron’s firing rate” Line 224: This sentence could be edited/reworded for clarity. Line 237: should “coherence” instead be “contrast” for this task? Line 299: “single-trial” -> “single-trials”


Review 4

Summary and Contributions: The paper develops a new method for the analysis of large scale brain recordings, to determine if different brain areas share task related information It introduces a relatively simple and circumscribed extension of previous work of Machens and colleagues (cited). The extension is much useful and needed. Having read the author's reply and the other comments, I stand by my original assessment.

Strengths: The problem to identify the task relevant information shared by different neural populations is fundamental for the progress of neuroscience. The authors make excellent inroads for providing tools to address this question.

Weaknesses: The work concentrates on linear methods. No reference is made to this limiation.

Correctness: yes, everything seems sound

Clarity: yes, very much so

Relation to Prior Work: Yes

Reproducibility: Yes

Additional Feedback: This is an excellent work and the following are rminor recommendations for further improvements. It would be useful if the authors acknowledge in their discussion the limitations and strengths of concentrating only on linear methods for their goal. Other techniques such as partial information decompositions have looked at this problem with nonlinear techniques, see e.g. Pica et al 2017 NeurIPS. It would be of interest if the author make readers aware of linear vs nonlinear approaches to this problem. It would be important that the code s shared in the final version of the article, at the moment is it announced as to be shared but it is not. The paragraph describing what is a pseudo-population vector is quite long but the concept is straightforward.

[Author Response · NeurIPS 2020]

Dear all reviewers: we highly appreciate your valuable comments and will reflect your comments in the revision.

**[R1] Interpretation of the shared component: ["...wouldn't two brain areas that encode a particular stimulus,**
**but not necessarily "communicate" with one another...exhibit a significant relationship? "]** We agree with the
reviewer that even if we find a significant relationship between two regions via dSCA, this does not guarantee that
they are communicating directly. As has been noted for other methods that seek to detect inter-regional connectivity
changes (e.g. Friston et al, Neuroimage 1997), there is always the possibility of a third region sharing information with
both source and target regions. This is one of the fundamental problems for analysing observational data in general,
and many methods have been proposed to detect causality (e.g., Shimizu et al., JMLR 2006). However, this caveat
would be true whether using averaged stimulus representations (as in our paper), or whether full trial-to-trial variability
(as suggested by the reviewer). The major advantage of using averaged representations is to allow for task-related
information to be marginalized in the analysis, demixing components that encode different task variables. Despite
this limitation, our method is an important starting point for subsequent interventional studies that more explicitly test
task-related communication in a causal manner. We agree with the reviewer that "task-related communication" is a bit
confusing, so we will change this to "task-related information sharing". We will also add these points in the Discussion.

**[R1] Why is the "Full information" source matrix useful?** The reviewer is correct: if the effects of different task
parameters on neural populations are independent, the results obtained from full information matrix and marginalized
matrix are indifferent. However, if their effects are not independent, marginalization by non-interesting task parameters
may unintentionally diminish the information of the task parameter of interest. Note that although this is also the case
when we use dPCA for a single brain region, this point was not discussed from this perspective. Thus, we view it as our
novel theoretical contribution. We will modify this sentence in the revision.

**[R2] Interpretability of the shared component.** This is important. In the right figure, we visu-
alize the contributions (absolute weights) to the first shared component for each neuron in the
simulation analysis (Fig. 2 in the current paper; also see Fig. 1e). dSCA correctly identified
neurons that contributed to (**a**) *Stimulus*- and (**b**) *Decision*-related communication. We emphasize
that dSCA's linearity makes it easy. Although we only analyzed the simulation data due to
insufficient space and time, we will add these results and results from real datasets to appendix.

**[R2] Novelty and originality.** We insufficiently emphasized the novelty and originality of dSCA.
Although several studies have investigated communication between populations of neurons [6-8],
task-related communication has been ignored. This is of fundamental importance in neuroscience,
and we show that it can be achieved simply by extending the previous method. While the reviewer
argues it is a comparatively simple solution rather than an entirely new computational approach,
we consider this a strength not a weakness because simplicity and ease of exposition are important
points for practitioners. In sum, we believe that our approach is a novel, original, and useful.

**[R2] Differentiation and advantages of dSCA.** Both dPCA and NMF has been applied to obtain linear decomposi-
tions. The important contribution of dSCA is that it tests how task-relevant information is *shared* across brain regions,
which is not the aim of dPCA or NMF. We will clarify this in the revision.

**[R2] Correlations among task variables.** We now realize that our explanation was insufficient. Although we only
focused on marginalizing a single parameter (e.g. stimulus or decision) in the current paper, we can also marginalize for
the interaction of multiple parameters, as was done in the dPCA paper [5]. We will clarify this in the revision.

**[R2-R4] Assumptions and limitations.** We agree that we should further describe the assumptions and limitations of
dSCA. dSCA assumes that task-related communication is linearly represented. It makes dSCA simple and exactly
solvable, and a linear method is popular in neuroscience because of its interpretability and less computational demand.
However, this is also the limitation of dSCA: it cannot capture non-linear communication. There have been several
methods for decomposing population neurons non-linearly, including deep learning. However, we are not aware
of studies applied to neuroscience data for investigating task-related communication between multiple regions in
low-dimensional projections of high-dimensional data. We believe our method is a good starting point for practitioners
and methodological exploration. We will clarify these two points in the Methods and Discussion sections.

**[R3] Regressing out before marginalization.** If two task parameters are orthogonalized by experimental design, we
do not need to do any procedure before marginalization (as in Fig. 4 in the current paper). However, if two task
parameters are correlated by design (as in Fig. 3), to focus on a task parameter, we need to regress the other task
parameters out from neural data before marginalization. We will clarify this point in the revision.

**[R3] Scaling of explained variance** We now realize that a more conventional definition of the explained variable is
subtracting these values from one. We will change this in the revision.

**[R1-R4] Other comments.** Though we cannot address all comments, we assure reviewers we will do so in the revision.

[Meta-Review · NeurIPS 2020]

Four knowledgeable referees support acceptance of this publication. The excitement about the novelty of the work is not shared across reviewers, however the reviewers agree that the work is correct and accurately addresses previous work. Given this, I agree that the paper should be accepted. Please make sure to address the clarification concerns pointed out by the reviewers in the final manuscript.